# Acquired and Transmitted Multidrug-Resistant Tuberculosis among the Incarcerated Population and Its Determinants in the State of Paraná—Brazil

**DOI:** 10.3390/ijerph192214895

**Published:** 2022-11-12

**Authors:** Márcio Souza dos Santos, Flávia Meneguetti Pieri, Thaís Zamboni Berra, Alessandro Rolim Scholze, Antônio Carlos Vieira Ramos, Juliane de Almeida Crispim, Clóvis Luciano Giacomet, Yan Mathias Alves, Fernanda Bruzadelli Paulino da Costa, Heriederson Sávio Dias Moura, Titilade Kehinde Ayandeyi Teibo, Jonas Bodini Alonso, Giselle Lima de Freitas, Ricardo Alexandre Arcêncio

**Affiliations:** 1Department of Maternal-Infant and Public Health Nursing, Ribeirão Preto College of Nursing, University of São Paulo, Ribeirão Preto 05403-000, SP, Brazil; 2Department of Nursing, State University of Londrina, Londrina 86039-440, PR, Brazil; 3Department of Nursing, Federal University of Minas Gerais, Belo Horizonte 30130-100, MG, Brazil

**Keywords:** tuberculosis, drug resistance, persons deprived of liberty, prisoners, public health

## Abstract

(1) Background: Tuberculosis remains a public health problem in the world. The study analyzed the factors associated with drug-resistant tuberculosis in the prison population of the state of Paraná. (2) Methods: Ecological study of drug-resistant tuberculosis cases registered in the Paraná Information System, Brazil (2008 to 2018). We performed descriptive statistics of quantitative parameters calculated with absolute frequencies. Additionally, we used binary regression logistics, where the odds ratio with its respective confidence interval was calculated. (3) Results: Of the 653 cases registered as cases of tuberculosis in the incarcerated population, 98 were drug-resistant tuberculosis. We observed that educational level of up to 8 to 11 years of schooling, negative bacterial culture (test outcome) and no tobacco use were factors associated with the non-development of drug-resistant tuberculosis, while clinically confirmed pulmonary TB and positive sputum smear microscopy in the fourth month of follow-up showed an association for the development of drug resistance. (4) Conclusions: The study showed that clinically confirmed pulmonary TB and a positive sputum smear microscopy in the fourth month of follow-up were associated with drug-resistant tuberculosis.

## 1. Introduction

Tuberculosis (TB) remains a serious public health problem worldwide due to its magnitude, persistence and strong social component, especially affecting people in vulnerable situations [1]. According to data from the World Health Organization (WHO), TB is one of the leading causes of death in the world, and until the arrival of the coronavirus, it was the leading cause of death from a single infectious agent, surpassing the human immunodeficiency virus (HIV)/acquired immunodeficiency syndrome (AIDS) [1].

In Brazil, in 2020, about 86,166 new cases of TB were reported; this represented a decrease compared to 2019 with a record of 96,655 cases of the disease. However, the decrease in the number of reported cases may be related to the context of the COVID-19 pandemic, in which there was a decreased search for new cases and, consequently, a reduced performance of laboratory tests and notification [2].

In Brazil, the Manual of Recommendations for the Control of Tuberculosis in Brazil establishes how the diagnosis of TB is performed, being subdivided mainly into clinical, differential, bacteriological, imaging and histopathological diagnoses [3].

The basic regimen used by Brazil to treat new cases in adults lasts for six months and consists of Rifampicin, isoniazid, Pyrazinamine and Ethambutol for two months and Rifampicin and isoniazid for the remaining four months [3].

In view of the strategies used to fight tuberculosis, vaccination stands out, which is established in the national vaccination schedule and at birth the child must be vaccinated against severe forms of TB [3].

The specific characteristics of drug-resistant tuberculosis (DRTB) is its resistance to any of the drugs considered for the treatment of TB when confirmed by the sensitivity test or GeneXpert MTB-RIF [3]. This constitutes a serious threat to public health, making control and elimination of the disease even more difficult [3].

DRTB is more frequent among the socially vulnerable groups, including the incarcerated population [4]. The incarcerated population is 28 times more likely to develop the disease when compared to the general population [4]. In recent years, the proportion of active TB cases in this population has increased significantly in Brazil, even surpassing the cases of TB-HIV co-infection [5].

The emergence of DRTB cases is largely related to the irregular use of prescribed drugs and lack of treatment [3]. According to the Brazilian national guidelines, TB treatment for those who have not been previously treated consists of rifampicin, ethambutol and isoniazid for at least six months [3]. Resistance to one of these first-line drugs requires replacement with second-line drugs and increased treatment time, which may predispose to dropping out of treatment or abandonment [3].

Treatment time for DRTB is three to four times longer compared to treatment time for sensitive cases; this often results in worse outcomes, such as treatment failures or treatment interruption [6,7,8].

In Brazil, the problem with TB eradication is not only in case detection, since most times ensuring that the detected cases on treatment complete their treatment is also as difficult. Following treatment, success is only around 71%, when the recommended rate is at least 85% [9]. This suggests that many people with TB are dropping out of treatment before completing it. The fact remains that the practice of abandoning treatment increases the possibility of developing DRTB.

In this context, it is worth mentioning the National Policy for Comprehensive Health Care for Persons Deprived of Liberty in the Penitentiary System (PNAISP), with the objective of expanding health actions of the Unified Health System (SUS) for this specific population [3].

The DRTB epidemic in the incarcerated population is concealed in many scenarios, which makes this an important and necessary topic to be investigated in order to advance knowledge in this area [10].

Thus, systematic monitoring of DRTB within prisons is essential to eliminating the possibility of community transmission. In addition, knowing which factors are associated with this clinical condition is strategic to designing broader intervention policies and projects. Thus, the study aims to analyze the factors that are associated with DRTB among the incarcerated population in the state of Paraná in Brazil.

## 2. Materials and Methods

### 2.1. Study Design and Location

This is an ecological study carried out in the state of Paraná, Brazil, considering the incarcerated population with sensitive TB and DRTB [11].

The study was carried out in the state of Paraná, one of the 27 federative units in Brazil, located in the southern region of Brazil. The state has a population of approximately 10 million inhabitants distributed across 399 municipalities, which represent 4.5% of the Brazilian population [12].

Regarding social indicators, the state of Paraná has the fifth highest Human Development Index (HDI) in the country (0.74), the fourth lowest illiteracy rate (0.52%) and the fourth lowest infant mortality rate (13.8 deaths/1000 live births) among the Brazilian federative units [13].

The prison system in Paraná is composed of 55 prison units distributed across nine regions of the State [14]. The latest survey presented by the National Penitentiary Department, carried out in 2020, points out that in Brazil the total number of the incarcerated population is 748,009 and the state of Paraná had 29,831, which represents 3.98% of the national population, 94.66% of this population is male, 62.20% is confined and 65.0% of them has at least one child. Regarding the vacancies available in prisons, there are 20,740 (95.03%) for men and 1084 for women (4.97%) [15].

### 2.2. Reference Population

The study population consisted of confirmed TB cases who were bacteriologically confirmed for DRTB registered in the Notifiable Diseases Information System (SINAN) between 2008 and 2018. The data was made available by the Paraná Secretariat of State Health, and to access the database, consent was sought from the coordinator of the State Tuberculosis Control Program.

The inclusion criteria for the study were being part of the incarcerated population and testing positive for TB. 

Based on the criteria established by the National Tuberculosis Control Program, after the detection of resistance, the State program for the fight against tuberculosis monitors the confirmed cases through SITE-TB. 

The exclusion criterion was cases in the notification system that contained blank data [15]. Generally, when it becomes infeasible to consider the entire population for research, it is conducted with sample calculation so that one can analyze a sample and make inferences about the whole.

In this research, the total number of cases reported by SINAN in the period from 2008 to 2018 in the incarcerated population corresponded to 2225 notifications, but in order for the study to become viable, cases in which the notification was not complete were excluded [15].

### 2.3. Statistical Analysis

First, we performed analysis of the database records and the descriptive analysis of presented cases, we used descriptive statistics and quantitative parameters to characterize the profile of the studied population, then absolute and relative frequencies were calculated through the use of IBM SPSS Statistics software version 25 (JMP Statistical Discovery, Canada).

To identify the factors associated with DRTB in the incarcerated population, we used binary logistic regression based on the variables present in the SINAN notification form. In the form, all the variables that explained the variable of interest were included. The dependent variable was DRTB.

The independent variables were: sex, age, race/color, education, type of entry (new case, recurrence, re-entry after abandonment, do not know, postmortem), clinical form, associated diseases (HIV, AIDS, alcoholism, diabetes, mental illness), use of illicit drugs, tobacco use, clinical examinations performed (chest radiography, culture, histopathology, bacilloscopy, sensitivity test or Molecular Rapid Test for Tuberculosis) and performance of directly observed treatment.

All selected independent variables were dichotomized into 0 and 1, (0: sensitive TB; 1: DR TB). In addition, the dependent variable (drug resistance) was also dichotomized.

The best model was chosen using a step-by-step selection method (Stepwise) that presented the lowest Akaike Information Criterion (AIC) [16,17]. Thus, the best model was chosen from the lowest values of the Akaike Information Criterion (AIC) [17].

After exhausting all possibilities of analysis and choosing the final model (smaller AIC), the likelihood ratio tests, we performed the Wald and McFadden tests to validate the model. It is also noteworthy that for the final model with the best comparison parameter, their odds ratio (OR) with their respective 95% Confidence Intervals (95% CI) were calculated for the statistically significant variables (*p* < 0.05).

Furthermore, the predictive ability and accuracy of the models were verified based on the area under the receiver operating characteristic (ROC) curve and their respective 95% CI values [18].

To identify the variables statistically related to DRTB, we used the Directed Acyclic Graph (DAG) through the RStudio program version 4.0.4 (Posit, Boston, MA, USA) and the Matchit package. Thus, to have the diagram postulated, we followed the steps below.

The first stage comprised the choice of variables from the regression logistic (culture, education, clinical form, smear and smoking). Homogeneity was evaluated using the chi-square test, followed by the adjustment of the score through complete pairing with the chosen variables using the chi-square test. After that, we analyzed the balance of the variables to see if the percentage of one group was similar to the other. At that moment, we observed the importance of including the variables age, sex and race to ensure the best balance between the variables.

It is noteworthy that the score adjustment was performed using the full matching method. Estimation was done using the mean treatment effect (ATE). The use of ATE is justified because it is an observational study, in which the population is unique.

Finally, we adjusted the regression model through complete correspondence and the DAG was built using the DAGitty computational tool (http://dagitty.net/dags.html, (accessed on 29 October 2022)). The DAG allowed us to observe the causal relationships between DRTB and the other independent variables.

### 2.4. Ethical Aspects

The study was authorized by the Paraná State Health Department—SESA and approved by the Ribeirão Preto School of Nursing with the Presentation Certificate for Ethical Assessment (CAAE) No. 31631520.2.0000.5393.3.

## 3. Results

In total, 653 TB cases were reported in the incarcerated population during the study period, with a minimum age of 18 years and a maximum of 82 years, the mean age was 27 years and the median age 29 years. Of the reported cases, 98 cases of DRTB were observed, with a minimum age of 18 years and a maximum of 61 years.

Table 1 presents the sociodemographic and clinical characterization of TB cases in the incarcerated population; males (n = 639; 97.9%), white race/color (n = 438; 67.1%), 5th to 8th grade incomplete (n = 256; 39.2%).

Table 2 also evidences that most cases had suspected TB (X-ray) (n = 504; 77.18%), sputum smear microscopy (n = 423; 64.78%) and positive sputum culture (n = 456; 69.83%), underwent directly observed treatment (n = 615; 94.18%) and showed resistance to the Isoniazid medication (n = 85; 13.02%), identified from the sensitivity test.

The “in progress” means that at the time the notification was made, the person deprived of liberty had performed the exam, however, they did not have the result.

Through logistic regression, it was possible to identify that having up to 8 and 11 years of education (OR: 0.41; 95% CI: 0.16–0.93), not using tobacco (OR: 0.02; 95% CI: 0.01–0.06) and negative sputum culture (OR: 0.29; 95% CI: 0.09–0.74) were negatively correlated with the outcome of interest. These factors give less chance for the development of DRTB.

The independent variables, the pulmonary clinical form (OR: 9.87, 95% CI: 1.55–23.81) and positive smear in the fourth month of follow-up (OR: 6.45, 95% CI: 1.04–53.79) were favorable for the development of DRTB. People with these characteristics and those of the incarcerated population are 9.87 and 6.45 times more likely to become ill with DRTB, respectively (Table 3).

For model validation, we used the pseudo R2 test (0.21), the Hosmer–Lemeshow test (*p* = 0.16), likelihood ratio (*p* < 0.01), CoxSnell (0.16), Nagelkerke (0.28) and McFadden (0.21). The model presented a value under the ROC of 0.79, classifying with the discriminatory power considered “sufficient”.

The authors recognize that the pulmonary clinical form and positive sputum smear in the 4th month of treatment had a large confidence interval; however, they emphasize that this was the best model for analysis, as well as suggesting that other studies be carried out to validate the findings presented.

Based on the DAG approach, Figure 1 shows the network of relationships between the variables, and thus, it is possible to identify the different links between them, allowing the assessment of the possibility of positive interventions to control and combat the occurrence of DRTB.

It is noted here that both clinical and social factors are related to the development of DRTB.

## 4. Discussion

This study evidenced the factors associated with TBDR in the incarcerated population in the state of Paraná. When conducting statistical analysis with logistic regression, we observed the variables with the highest and lowest chance of developing DRTB. Furthermore, from the postulation of the diagram, it was possible to assess the relationship between the independent variables and the outcome of DRTB.

When evaluating the level of education, it was found that the higher the level of education, the lower the chance of developing DRTB as the level of education directly reflects on self-care with health. Low education can make the individual more subject to risky behaviors and reduce their perception of ideas and self-care in relation to their clinical condition, this is associated with possible treatment failure, thus allowing the possibility of drug resistance, abandonment and death from TB [19].

From this perspective, low schooling (less than eight years of schooling) has been associated with a higher risk of treatment abandonment during treatment for DRTB, and this is in turn associated with a set of precarious socioeconomic conditions [20,21].

The literature highlights the unhealthy habits and lifestyle of this population, potentiating the factors that increase the risk of contracting TB, such as the use of illicit drugs, alcohol, tobacco, malnutrition and even other pathologies, which corroborates the findings of this study [22].

The relationship between using illicit drugs and having TB has been increasing, evidencing a public health problem [23]. Research carried out at a university hospital in São Paulo showed that patients who consume alcohol, are smokers and are users of illicit drugs fail to undergo TB treatment more frequently than those who do not have any of these risky lifestyles [23]; this increases the probability of progression to drug resistance for TB.

The pulmonary clinical form was the most prevalent, and it can be assumed that it is the form that most causes drug resistance. However, considering that in the cases included in this study, 98.98% had this clinical form, it is expected that most cases of drug resistance will present the pulmonary clinical form.

With regard to smear microscopy, it is known that it is a simple, safe, low-cost method used all over the world [24]. In view of the above, it is expected that those who have a negative sputum culture are less likely to develop DRTB. This is because having a positive smear in the fourth month of treatment represented a threat, as it may indicate that the drugs used did not have the expected effect and, therefore, does not interrupt the chain of transmission and increases the chance of developing DRTB.

Another important factor for reflection refers to the incarcerated population itself since the Brazilian prison system was regulated in 1984 there has been debates on fundamental rights [21]. From this perspective, the increase in the number of people deprived of liberty is a reality in Brazil, since in December 2017 the country had the third largest prison population in the world, with a predominance of blacks having low education, out of which 88% had not completed high school [21].

The structural conditions in prisons are detrimental for the health-disease process of the incarcerated population. The environment is hostile and unhealthy, and this allows for the occurrence and spread of several diseases, including TB [25].

This reality is also present in other countries, since TB in the incarcerated population is a global problem, especially in developing countries. Unhealthy conditions, inadequate treatment, poor ventilation, overcrowding, lack of sun with consequent vitamin D deficiency, among other aspects, contribute negatively to the spread and permanence of the disease in this population [10,26].

The inadequate treatment of TB carried out inside the prisons exposes other people who live with the incarcerated population, both the workers and the visitors themselves, which facilitates the lengthening of the transmission chain [10].

Thus, in addition to enhancing the transmission of the disease, it increases the possibility of transmission of TB in its resistant form. This ends up making treatment and consequently disease control difficult, making it impossible to achieve the third objective of sustainable development, article 3.3 of the 2030 agenda proposed by the United Nations, with aims to end epidemics, including TB [10,27].

With the growth of this population, the implementation of the National Tuberculosis Program policy in prisons becomes increasingly important, since the actions proposed by the Ministry of Health are partially applied [21,28].

TB is a disease that can be treated and cured; however, the patient must commit to the treatment until the end, as well as the State guaranteeing effective treatment through directly observed treatment [3,7,8,10,22]. Thus, one of the likely reasons for high TB mortality rate, incidence and emergence of multidrug-resistant bacilli would be the patients’ lack of adherence to treatment [4,26].

Delay in the diagnosis of the disease is related to the naturalization of the lack of care for the incarcerated population, the interpretation that the prison is a place of “death” and “suffering” and the deprivation of the right to health for the incarcerated population due to their position in society [29,30]. All of these highlights the inequity in access to health care for this population group [29,30]. Thus, this context points to another major challenge in disease control, which is the need for changes in general perception about the right to health in prisons [22].

The invisibility of this population by the state, little concern with resocialization because it is a neglected population, low income and schooling, poverty, precarious conditions, overcrowding and prejudice against this population is the reason why TB and other diseases prevail in the prison system [25].

From the DAG, it is possible to look back at the relationships that the variables establish with each other and although some relationships are expected, the performance of the smear in the fourth month, for example, is a finding that allows us to reflect on the importance of monitoring TB especially in the population of this study.

Considering that the study indicated that the incarcerated population with positive sputum smear in the fourth month of treatment is a factor associated with DRTB and that the habit of not smoking and the presence of a negative culture presents a lower chance of developing DRTB, it is essential to develop public service policies aimed towards this population. We suggest investment in improving the health conditions of the prison system, healthy habits/behaviors promotion and monitoring of symptomatic respiratory diseases.

It is noteworthy that the study has limitations related to the data collected from the SINAN of the State of Paraná, since the data may suffer underreporting, mainly due to the diagnostic difficulty for DRTB. This is caused, for example, by the lack of indication of patients for tests of sensitivity, as well as the difficulty in collecting information in a database and the impossibility of knowing and evaluating the history of previous treatment for TB.

It is also noteworthy that the development of this study allows the evaluation of DRTB in the penitentiary system, which will provide the health services and management with a reflection on the actions developed in favor of the treatment, control and monitoring of this disease. This supports health managers in the restructuring of disease surveillance strategies, making it possible to discuss the relationship between the determinants of drug resistance and tuberculosis, which is still an important knowledge gap.

The authors recognize the importance of the subject and encourage and recommend other publications. We recommend the importance of continuing with the study, so that the geo-referencing of these cases can be presented, to show where the cases represented are on the map, in addition to presenting how TBDR will “behave” in the coming years and other analyses that provide subsidies for improving the process of health-disease.

## 5. Conclusions

From the results, factors associated with DRTB were evidenced as public health problems that contribute to the permanence of this disease. There is a need for advances to be made to improve the public health network from the perspective of tuberculosis surveillance, so that there is continuous improvement of health surveillance in the country. Actions such as the systematic use of smear microscopy throughout treatment are an important predictor to predict the TBDR occurrences.

We highlighted the importance of building and applying viable policies consistent with the reality of penitentiaries, considering all aspects that permeate it, without forgetting the inequity of access to health due to the naturalization of neglect, indifference and prejudice instituted by the society.

## Figures and Tables

**Figure 1 ijerph-19-14895-f001:**
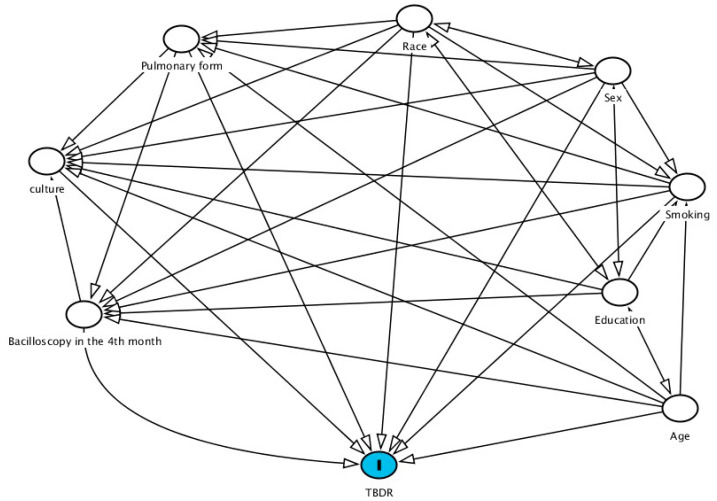
Directed acyclic graph, representing the hypotheses about the relationships between DRTB and the independent variables in the incarcerated population between 2008 and 2018 in the State of Paraná, Brazil.

**Table 1 ijerph-19-14895-t001:** Sociodemographic and clinical characterization of tuberculosis cases in the incarcerated population reported between 2008 and 2018 in the State of Paraná, Brazil.

	TBDR (n = 98)		TB- SENSITIVE (n = 555)		TOTAL (n = 653)	
Variables	n	%	n	%	n	%
Age (Years old)						
18–29	64	65.31	329	59.28	393	60.18
30–59	33	33.67	220	39.64	253	38.74
60 or more	1	1.02	6	1.08	7	1.07
Sex						
Masculine	97	98.98	542	97.66	639	97.86
Feminine	1	1.02	13	2.34	14	2.14
Race						
White	58	59.18	380	68.47	438	67.08
Black	14	14.29	39	7.03	53	8.12
Yellow	0	0.00	3	0.54	3	0.46
Brown	25	25.51	130	23.42	155	23.74
Indigenous	0	0.00	1	0.18	1	0.15
Ignored	1	1.02	2	0.36	3	0.46
Education						
No education	1	1.02	9	1.62	10	1.53
1st to 4th grade incomplete	9	9.18	48	8.65	57	8.73
4th grade complete	7	7.14	53	9.55	60	9.19
5th to 8th grade incomplete	37	37.76	219	39.46	256	39.20
Complete primary education	11	11.22	64	11.53	75	11.49
Incomplete high school	6	6.12	51	9.19	57	8.73
Complete high school	4	4.08	33	5.95	37	5.67
Incomplete higher education	0	0.00	2	0.36	2	0.31
Complete higher education	0	0.00	1	0.18	1	0.15
Ignored	23	23.47	75	13.51	98	15.01
Input Type						
New case	80	81.63	483	87.03	563	86.22
Relapse	10	10.20	41	7.39	51	7.81
Re-entry after abandonment	7	7.14	28	5.05	35	5.36
Do not know	1	1.02	1	0.18	2	0.31
Post-death	0	0.00	2	0.36	2	0.31
Clinical Form						
Pulmonary	97	98.98	514	92.61	611	93.57
Extrapulmonary	0	0.00	24	4.32	24	3.68
Pulmonary + Extrapulmonary	1	1.02	17	3.06	18	2.76

**Table 2 ijerph-19-14895-t002:** Risk factors and characteristics regarding the diagnosis of tuberculosis in adults deprived of liberty reported between 2008 and 2018 in the State of Paraná, Brazil.

	TBDR (n = 98)		TB- SENSITIVE (n = 555)		N TOTAL (n = 653)	
Variables	n	%	n	%	n	%
AIDS						
Yes	3	3.06	32	5.77	35	5.36
No	86	87.76	499	89.91	585	89.59
Ignored	9	9.18	24	4.32	33	5.05
Alcoholism						
Yes	14	14.29	71	12.79	85	13.02
No	74	75.51	457	82.34	531	81.32
Ignored	10	10.20	27	4.86	37	5.67
Diabetes						
Yes	1	1.02	9	1.62	10	1.53
No	88	89.80	518	93.33	606	92.80
Ignored	9	9.18	28	5.05	37	5.67
Mental disease						
Yes	1	1.02	5	0.90	6	0.92
No	88	89.80	524	94.41	612	93.72
Ignored	9	9.18	26	4.68	35	5.36
Illicit drugs						
Yes	40	40.82	279	50.27	319	48.85
No	38	38.78	266	47.93	304	46.55
Ignored	20	20.41	10	1.80	30	4.59
Smoking						
Yes	43	43.88	285	51.35	328	50.23
No	37	37.76	263	47.39	300	45.94
Ignored	18	18.37	7	1.26	25	3.83
X-ray						
Suspect	70	71.43	434	78.20	504	77.18
Normal	3	3.06	11	1.98	14	2.15
Unrealized	25	25.51	109	19.64	134	20.52
Ignored	0	0.00	1	0.18	1	0.15
Sputum Culture						
Positive	84	85.71	372	67.03	456	69.83
Negative	5	5.10	85	15.31	90	13.78
In progress	3	3.06	12	2.16	15	2.30
Not performed	6	6.11	86	15.50	92	14.09
HIV						
Positive	3	3.06	32	5.77	35	5.36
Negative	93	94.90	485	87.38	578	88.52
In progress	1	1.02	3	0.54	4	0.61
Not performed	1	1.02	35	6.30	36	5.52
Histopathology						
Alcohol-Acid Resistant Bacilli Positive	3	3.06	23	4.14	26	3.98
Suggestive of TB	2	2.04	14	2.52	16	2.45
Not suggestive of TB	0	0.00	1	0.18	1	0.15
In progress	0	0.00	4	0.72	4	0.61
Unrealized	89	90.82	505	90.99	594	90.96
Ignored	4	4.08	8	1.44	12	1.84
Immigrant						
Yes	0	0.00	3	0.54	3	0.46
No	81	82.65	550	99.10	631	96.63
Ignored	17	17.35	2	0.36	19	2.91
Receive government benefit						
Yes	1	1.02	10	1.80	11	1.68
No	75	76.53	525	94.59	598	91.58
Ignored	22	22.45	20	3.60	44	6.74
Rapid molecular test						
Sensitive to Rifampicin	43	43.88	428	77.12	471	72.13
Resistant to Rifampicin	18	18.37	6	1.08	24	3.68
Undetectable	1	1.02	47	8.47	48	7.35
Ignored	22	20.41	20	3.60	44	6.74
Unrealized	16	16.33	1	0.18	17	2.60
Sensitivity test						
Resistant to Isoniazid only	53	54.08	32	5.77	85	13.02
Resistant only to Rifampicin	3	3.06	0	0.00	3	0.46
Resistant to Isoniazid and Rifampicin	4	4.08	1	0.18	5	0.77
Resistant to other first-line drugs	5	5.10	2	0.36	7	1.07
Sensitive	8	8.16	304	54.77	312	47.78
In progress	0	0.00	10	1.80	10	1.53
Unrealized	5	5.10	55	9.91	60	9.19
Ignored	20	20.41	151	27.21	171	26.19
Antiretroviral therapy						
Positive	1	1.02	28	5.05	29	4.44
Negative	93	94.90	493	88.83	586	89.74
Ignored	4	4.08	34	6.13	38	5.82
Transfer						
Different state	2	2.04	10	1.80	12	1.84

**Table 3 ijerph-19-14895-t003:** Explanatory model for drug-resistant tuberculosis in the incarcerated population in the state of Paraná, Brazil.

	Coefficient	Pr (>|z|)	Odds Ratio (95% CI)
Education: 8 to 11 years of study	−0.87	0.04	0.41 (0.16–0.93)
Clinical form: Pulmonary	2.28	0.05	9.87 (1.55–23.81)
Not use of tobacco	−3.84	<0.01	0.02 (0.01–0.06)
Negative sputum culture	−1.20	0.01	0.29 (0.09–0.74)
Positive bacilloscopy in the 4th month of treatment	1.86	0.05	6.45 (1.04–53.79)

## Data Availability

The data presented in this study are available from the corresponding author on reasonable request.

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
