# Peer review of "Acquired and Transmitted Multidrug-Resistant Tuberculosis among the Incarcerated Population and Its Determinants in the State of Paraná—Brazil"

_ijerph, 2022, doi:10.3390/ijerph192214895_

Round 1

Reviewer 1 Report

Souza dos Santos et al present their findings on risk factors associated with infection with drug resistant TB in a Brazilian prison population. 

The subject of the paper is relevant and important and would be interesting to a variety of readership, but some concerns exist with data categorization (and consequently validity of the analysis):

Table 2 provides percentages of resistance in "TB-Sensitive" group (in the "sensitivity" part of the table, e.g. 5.77% INH-monoR, 0.18% MDR, 0.36% R to other first line drugs).  It's not clear whether this is a mistake in data presentation in the table (and how this mistake came to be, if that's the case), or a mistake in the way cases were categorized. If resistant cases were included in the "TB Sensitive" group, the analyses are not valid and need to be redone. 

Some minor comments:

1. Paper should be edited throughout for language and clarity (e.g. what are "unrealized" and "ignored" groups? what is a "do not know" input type? some words are used incorrectly - e.g. "transcendence" in Introduction, "breed" to describe "race"), as well as punctuation (random periods in several parts of the paper)

2. Definitions for some acronyms are missing (TRM-TB, BAAR)

3. Referencing within text is inconsistent (no parenthesis used for references in several instances)

4. "Table 3" is referenced but is missing from the manuscript

5. Abstract erroneously states that higher education was associated with resistant TB infection, contrary to what is concluded in the body of the paper

6. Argument is made that positive smear microscopy in the 4th month of treatment is an indication of drug ineffectiveness, which is a contentious statement - cavitary TB cases can have prolonged smear positivity in the absence of positive cultures, due to ongoing expectoration of dead bacilli. 

7. Inclusion criteria erroneously state that the patients needed to have both a positive TB test and confirmed resistance (lines 111-113), but clearly drug sensitive cases were the control group

8. Footnote for tables 1 and 2 says "source: author himself" - I'm not clear on what is meant, but it's doubtful that an individual could be the source of the type of data presented in these tables. 

Author Response

Dear Reviewer,

Thank you for reviewing the manuscript “Acquired and transmitted multidrug-resistant tuberculosis among the incarcerated population and its determinants in the State of Paraná – Brazil”.

We would like to thank the reviewers for their helpful and comprehensive comments, which clearly contributed to improving our study.

As requested, we present our responses to each point raised by the reviewers. Changes made to the manuscript were highlighted in red using the word tool, control changes.

Yours sincerely,

Márcio Souza dos Santos and co-authors.

Reviewer 2 Report

Tuberculosis is still an important epidemiological, medical social and social environment of the modern world. According to data from the World Health Organization, in 2015 10.4 million people worldwide suffered from tuberculosis. 1.4 million of HIV-negative people and 400,000 died of this disease among HIV-positive ones. The incidence of tuberculosis in 2015 was 142 / 100,000 worldwide. Patients with co-existing HIV infection pose a special challenge to modern medicine. Multi-drug resistant strains of M. tuberculosis (MDR) pose a serious therapeutic problem. The presence of such a resistance mechanism limits the already short list of antituberculosis drugs. In extreme cases, strains with extended XDR drug resistance are selected, for which there is often no therapeutic option. Patients infected with such strains constitute a reservoir of mycobacteria with which they can infect other people from their surroundings. It is a dangerous phenomenon, especially within closed environments - camps or prisons. This led to a huge spread of these strains in many economically underdeveloped countries.

The above manuscript by dos Santos et al. will fit  in the current diagnostic trends very well. During the SARS-CoV-2 pandemic, the number of patients who had symptoms of tuberculosis was asleep dramatically. Access to specialist clinicists was also difficult, and thus the number of diagnostic materials dropped, in some cases even to zero. The patients were left on their own, with no possibility of continuing or implementing therapeutic treatment. From the statistical point of view, the incidence of tuberculosis has decreased significantly at that time. However, this was not the result of any increase in the effectiveness of treatment, and the lack of therapeutic and diagnostic possibilities during the pandemic as well. All this is very accurately proved by the facts presented by the authors, based on the State of Parana in Brazil. The authors proved their theses in an understandable and logical way. They resisted the large population of the country, mainly among those in prison and from the margins of society.

The only thing missing in my opinion is the diagnostic protocol used in Brazil. Perhaps its simplified version would increase the quality of the article. Each country has different guidelines for dealing with Mycobacterium tuberculosis and confirms cases of tuberculosis. If the authors were to briefly describe such a scheme, it would enhance the value of the proposed manuscript.

Author Response

(The authors gave the same response as above.)
